# The Perception of the Quality of Professional Healthcare Assistance for the Management of Endometriosis: Findings from a National Survey in Italy

**DOI:** 10.3390/ijerph20216978

**Published:** 2023-10-26

**Authors:** Vincenza Cofini, Mario Muselli, Erika Limoncin, Chiara Lolli, Erika Pelaccia, Maurizio Guido, Leila Fabiani, Stefano Necozione

**Affiliations:** 1Department of Life, Health and Environmental Sciences, University of L’Aquila, 67100 L’Aquila, Italy; vincenza.cofini@univaq.it (V.C.); chiara.lolli@uniroma1.it (C.L.); erika.pelaccia@graduate.univaq.it (E.P.); maurizio.guido@univaq.it (M.G.); leila.fabiani@univaq.it (L.F.); stefano.necozione@univaq.it (S.N.); 2Department of Dynamic, Clinical Psychology and Health Sciences, Sapienza University of Rome, 00185 Rome, Italy; erika.limoncin@uniroma1.it

**Keywords:** quality of care (QoC) satisfaction, endometriosis, patient’s healthcare quality perception, patient satisfaction

## Abstract

(1) Background: endometriosis is included in the list of chronic and disabling pathologies. This study aimed to examine patients’ points of view about the quality of care for endometriosis during the COVID-19 pandemic; (2) Methods: we conducted a survey on knowledge about endometriosis, management of endometriosis, perceived mental and physical well-being, and perceived changes in the quality of care during the pandemic; (3) Results: out of 1065 participants, 875 were included in the analysis, with an average age of 34. Overall, patients had positive perceptions of care accessibility and cleanliness (95%), but less satisfaction with visit hours (86%). Those with better physical and psychological health were more satisfied with service hours, while those treated at specialised centres were more content with cleanliness. Satisfaction with clarity was linked to specialist treatment, and perceived availability to listen correlated with age, complications, and overall health status; (4) Conclusions: patients’ perspectives are crucial for patient education and advocate for specialised interdisciplinary networks to support endometriosis management and patients’ well-being. These findings highlight the importance of considering patient viewpoints, particularly in the context of the COVID-19 pandemic’s impact on healthcare systems and suggest a need for further research from the patient’s perspective.

## 1. Introduction

Endometriosis is a condition in which tissues, such as the inner lining of the uterus (endometrium), grow outside the uterus. This tissue can grow in any body part but most commonly affects the ovaries, fallopian tubes, and tissues lining the pelvis [1]. Notably, the worldwide prevalence rate of endometriosis is estimated to be between 6 and 10% in the general female population [2] and 70% in women with chronic pelvic pain [3].

This disease is most frequently diagnosed in women of childbearing age. However, it can take a long time to obtain a diagnosis (on average, 11 years in the USA and 8 years in the UK) because of the disorder’s variable symptoms and indicators, which can also cause it to be mistaken for other disorders [4].

According to the Italian Ministry of Health, Italian women diagnosed with endometriosis number over three million, with a 10–15% prevalence rate among women of reproductive age. Unfortunately, many of these women frequently experience intense pain and a poorer quality of life (QoL) due to a delay in endometriosis diagnosis. The Ministry of Health reported an average delay of 7 years in 2021 and 10.7 by Brandes et al. 10.7 years in 2022 [5].

In 2006, the Italian Senate Committee investigated the cost of hospital admissions for endometriosis in the Italian National Health System, estimating it to be 2773.80 euros per patient, totalling 54 million euros annually. To address this problem, the Committee proposed a 5-year plan that included the following measures: identifying disease indicators with a dedicated diagnosis-related group for reimbursement, collaboration between relevant ministries and institutes, establishing a care network with specialised centres, allocating funds, providing free medical therapies for chronic cases, assigning disability status for severe cases under Act no. 104 of 5 February 1992, and promoting information and awareness campaigns [6]. Starting in 2017, endometriosis in Italy was introduced into the Essential Levels of Assistance (LEA) and included in the list of chronic and disabling pathologies in the most advanced clinical stages (“moderate or III degree” and “severe or IV degree”). Patients in these stages have the right to benefit from certain specialist check-up services with exemption. Approximately 300,000 exemptions are estimated by the Ministry of Health. Endometriosis, being a chronic disease without a cure yet, is primarily treated by gynaecologists, but it requires long-term care [7,8,9].

The Chronic Care Model (CCM) is a comprehensive framework for providing high-quality treatment aiming to improve QoL in patients with chronic diseases. With its essential tenets of encouraging self-management of care, the CCM may serve as a respectable substitute for health management [10]. It may be used to treat endometriosis in women, and it is possible to establish a long-term, all-encompassing, multidisciplinary program that is patient-focused [9]. In line with the CCM evidence, it is suggested to be considered as an endpoint and to evaluate health-related QoL (HR-QoL) in patients suffering from endometriosis. Notably, HR-QoL includes both physical and mental QoL. Several studies have demonstrated the negative repercussions of endometriosis on both physical (pain, [11]) and mental (delay in diagnosis [11,12], body perception, body image, sexual functioning, and body compassion [13,14,15]) QoL. As reported by Yoldemir [16], the Health Questionnaire SF-36 V2 Standard (SF-36) is a valid tool for evaluating the quality of general health.

Hence, based on the literature, to ensure that patients receive the best possible treatment, it is crucial to assess the presence of endometriosis as soon as possible and to start with multidisciplinary treatment to improve the quality of healthcare services. According to the Committee of Ministers to Member States on the Development and Implementation of Quality Improvement Systems (QIS) in Healthcare, it is important to consider outcomes regarding patient health, well-being, and satisfaction to improve healthcare quality at all levels [17]. As a healthcare quality indicator, patient satisfaction is highly relevant for chronic diseases [18,19,20,21]. Specifically, quality of care and integrated management of chronic diseases in the territory are considered fundamental conditions for effectively preventing complications and improving patients’ quality of life [22,23].

As for the Italian context, most of the studies on managing patients with chronic conditions are limited to chronic diseases [10]. Furthermore, the COVID-19 pandemic had a strong impact on healthcare, and after the lockdown, the Italian health system had to replan all health services, the management of diseases, their clinical and surgical treatments, and follow-up [24,25,26]. Similarly, the COVID-19 pandemic has also impacted endometriosis care and quality of life [27,28].

The present study aimed to analyse patients’ points of view regarding the quality of care for endometriosis during the COVID-19 pandemic. To the best of our knowledge, no studies have been conducted in Italy on endometriosis healthcare satisfaction from a patient’s perspective during the COVID-19 pandemic. In particular, this study’s objective was to evaluate the quality of assistance received during the COVID-19 pandemic and its association with the management of endometriosis, perceived mental and physical well-being, and perceived changes in assistance due to the COVID-19 pandemic.

## 2. Materials and Methods

We conducted a secondary analysis of data from a cross-sectional study authorised by the Internal Review Board (IRB) of the University of L’Aquila (Protocol number 26/2021) and carried out from July to September 2021 on Italian women who self-reported a diagnosis of endometriosis. This project had the following objectives: estimate disease knowledge, sociodemographic and clinical characteristics of the investigated population, quality of life, healthcare satisfaction, and the COVID-19 impact perceived on healthcare. This manuscript reports an analysis of healthcare satisfaction and its associated factors, particularly considering the following outcomes:➔perceived quality of healthcare (Organizational, Environmental, and Relational quality).➔knowledge about endometriosis (disease duration, time of diagnosis).➔management of endometriosis (therapeutic and surgical treatments).➔perceived mental and physical well-being.➔perceived changes in quality of care due to the COVID-19 pandemic.

### 2.1. Participants and Recruitments

The research objectives were presented with an informative note which also reported the team’s names and contacts (email and telephone numbers) to ask for details and explanations of the project. Administrators of online groups of Italian women with endometriosis and members of endometriosis associations registered on social networks such as Facebook and Instagram were contacted before questionnaire administration. Dissemination and awareness of participation were further enhanced by three public figures on Instagram.

The inclusion criteria were being a woman living in Italy, having been diagnosed with endometriosis by a gynaecologist, and understanding Italian. Participants were guaranteed anonymity and respect for privacy, and they could fill out the questionnaire only if they provided informed consent and data processing authorisation.

For the present study, as part of a large survey, we analysed data from participants who reported having had a medical (gynaecological) visit for endometriosis during the last 12 months before the interview because of the reference period of the questionnaire used [29]. Data concerning pregnant women (*n* = 20) were excluded from the analysis because pregnancy may be associated with an altered quality of life [7] and medical visits may be related to pregnancy.

### 2.2. Tools

A self-report questionnaire on Google Forms was used in this survey. It included items to provide sociodemographic and lifestyle data (smoking habit, alcohol consumption, sedentary status) and clinical information about body mass index (BMI), time of endometriosis diagnosis, pregnancy, treatment, complications related to endometriosis, and comorbidities. The Italian version of the Health Questionnaire SF-36 V2 Standard was used to assess perceived health status; this is a standardized questionnaire that assesses physical and mental health articulated into 36 questions to evaluate eight subscales (domains) and two standardised components: the physical component summary (PCS) and the mental component summary (MCS). Higher scores indicated a better quality of life [30,31].

Satisfaction with healthcare was assessed through a questionnaire used in other studies to investigate the perception of Organizational Quality (appointment wait times, waiting time from arrival to a medical visit, and appointment scheduled), environmental quality (visit hours, accessibility, and cleanliness), Relational quality (professionals’ courtesy, clarity of information, and listening), an item to evaluate Cooperation Quality among professionals, and overall rating of the service offered in the last 12 months [21,29]. Furthermore, we asked the participants if they perceived changes between the last visit and the previous visits that took place during the pre-COVID-19 period, specifically regarding Environmental and Relational quality (neutral/positive/negative); subsequently, we inquired if they perceived that these changes were related to the COVID-19 pandemic.

### 2.3. Sample Size

Snowball sampling was performed via email, social media networks, and instant messaging applications. The sample size was estimated at 1032 units, considering the large population, a precision of ±3% with a 95% confidence interval, and a response level for a single parameter equal to 50% [32]. A logistic regression of a binary response variable (Y) on a continuous normally distributed variable (X) with a sample size of 617 observations achieves 95% power at a 0.05-significance level to detect a change in Prob(Y = 1) from the value of 0.2 at the mean of X to 0.27273 when X is increased to one standard deviation above the mean. This change corresponds to an odds ratio of 1.5. An adjustment was made since a multiple regression of the independent variable of interest on the other independent variables in the logistic regression obtained an R-Squared of 0.2 [33].

### 2.4. Statistical Analysis

All data were analysed using descriptive statistics and expressed as percentages and frequencies or mean and standard deviation (SD) for categorical or numerical data. Each item related to the following dimensions: environment (Hours, Accessibility, and Cleaning), relational (Courtesy, Clarity of Information, and Willingness to Listen), and cooperation (Level of Cooperation) satisfaction. These dimensions were dichotomised into positive and negative perceptions and were then considered dependent variables for separate multivariable logistic regression models. In each multivariable model, the independent variables included age, residence (divided into southern, central, and northern Italy), employment status (yes/no), marital status (married-cohabiting/single), educational level (high: degree or above/low: secondary school), health service provider (specialised/non-specialised for endometriosis), complications related to endometriosis (yes/no, including at least one complication among pelvic pain, dyspareunia, pelvic floor disorder, self-catheterism, neuropathic disorder, infertility, hysterectomy, salpingectomy, ovariectomy, intestinal stenosis, intestinal resection, bladder resection, adherence, and others), comorbidities (yes/no, answered to the question “do you suffer from other diseases?”), and the Physical and Psychological summary components of the SF-36 (PCS and MCS). Additionally, for the environmental and relational dimensions, the models considered changes perceived due to the COVID-19 pandemic (neutral/positive/negative). Adjusted odds ratios (AOR) and confidential intervals (95%CI) were reported.

## 3. Results

A total of 1065 women participated in the survey. The results reported in the present study refer to 875 of 1065 women who had undergone a medical visit for endometriosis at least 12 months prior to the interview. The mean age of the patients was approximately 34 years (SD = 7.6) with a range of 18 to 55 years. Among them, 68, 320, and 187 participants were from Northern Italy, Southern Italy (including the islands), and Central Italy, respectively. Additionally, 86 (9.8%) participants reported low levels of education. Based on the weight and height data, we computed the BMI of the participants. As shown in Table 1, 26% of them fell into the overweight or obese category. Furthermore, over half (58%) reported being either inactive or having never engaged in physical activity, hence they were classified as sedentary. Regarding their knowledge about their disease, 397 participants (45%) stated that they became aware of their condition four years prior to the survey completion. Furthermore, 21% of the participants (187) reported that they only took medicines if prescribed or recommended by gynaecologists, while 41% (360) admitted to using painkillers without specialist recommendation. Of the respondents, 687 (79%) had experienced complications related to endometriosis, specifically dyspareunia (68%, 598/875) and chronic pelvic pain (65%, 570/875). More than half of the participants declared having at least one other disease (55%). The other diseases reported with greater frequency included endocrinological disorders or pathologies (15%, 134/875), autoimmune or rheumatological pathologies (11%, 97/875), and chronic pelvic pain (65%, 570/875).

The diagnostic delay exceeded 9 years for 35% of the patients (303). Regarding health status, the summary component of the SF-36 questionnaire indicated that the participants perceived their physical well-being to be better than their mental well-being, as detailed in Table 1.

Furthermore, 79% reported that they were treated by a centre or a doctor specialising in endometriosis treatment. As shown in Table 2, 88% of the women reported that their waiting time for a visit was less than six months, and only 20% (one woman out of five) reported having a scheduled appointment during the visit. During the last 12 months, some women had consulted other healthcare professionals for endometriotic complications, wherein approximately half of them (45%) had consulted a nutritionist.

Women who visited in the last year were asked questions about the quality of healthcare services and the healthcare workers they encountered during the last visit. As reported in Table 3, the prevalence rate of positive perceptions about environmental quality was 95% for Accessibility and Cleaning (831/875 and 828/875, respectively), and the proportion of positive perceptions about visit hours was 86% (751/975). The multivariate analysis for each dimension of environmental quality showed that women with physical and psychological wellness were more satisfied with hours of health service, and the associations between perceived quality and health status were significant (PCS: AOR = 1.02; 95%CI: 1.00–1.04; *p* = 0.034; MCS: AOR: 1.03; 95%CI: 1.01–1.05; *p* = 0.001). Concerning accessibility, the PCS component was the only factor associated, with an AOR = 1.05 (95%CI: 1.01–1.09; *p* = 0.006]. Furthermore, women treated by a centre or doctor specialised in endometriosis treatment were more satisfied with the cleaning than women treated by other health providers (AOR = 3.46; 95%CI: 1.85–6.48; *p* < 0.001). Notably, the satisfaction of the quality of cleaning was also significantly associated to the presence of complications. In fact, women with complications were less satisfied than women without complications (AOR = 0.32; 95%CI: 0.12–0.85; *p* = 0.022).

With respect to the positive perception of relational quality, as reported in Table 4, 748/875 women were positively satisfied with professional courtesy, 707/875 with the clarity of information, and 640/875 with the availability to listen. Women with better health status were more satisfied with the curriculum of health professionals. In fact, PCS and MCS were significantly associated with positive judgments: PCS, AOR = 1.03, *p* = 0.015, 95%CI: 1.01–10.5; and MCS, AOR = 1.04, *p* < 0.001, 95%CI: 1.02–1.06, respectively. Satisfaction with the clarity of medical information and recommendations was associated with the type of provider selected to treat the disease. Women treated by specialists in endometriosis were more satisfied (AOR = 1.80; *p* = 0.004; 95%CI: 1.20–2.69). The positive perception of clarity was also reported by women with a better physical and mental health status [PCS: AOR = 1.04; *p* < 0.001; 95%CI: 1.03–1.06; MCS: 1.04, *p*< 0.001; 95%CI: 1.02–1.05]. A positive perception of the quality of the availability of listening was associated with age, presence of complications, and health status. Older women were more satisfied than younger women, AOR = 1.02; *p* = 0.037; 95%CI: 1.00–1.05, and women with complications were also more satisfied than women without complications, reporting AOR = 1.46, *p* = 0.044; 95%CI: 1.01–2.11. Furthermore, higher levels of PCS and MCS were also significantly associated with positive perceptions.

Furthermore, positive satisfaction with cooperation and multidisciplinary organization, as depicted in Table 5, was perceived by 480/875 (55%) participants and was associated with educational level whereby it was lower in women with high levels (AOR = 0.62; 95% CI: 0.46–0.84; *p* = 0.002). The positive perception of the quality was related to the type of health service, being higher in specialised centres (AOR = 1.82; 95% CI: 1.30–2.63; *p* = 0.001), and to the two components of the SF-36 test (PCS and MCS).

Regarding aspects of organizational, environmental, and relational quality, 57% of participants (*n* = 500) reported changes related to the COVID-19 pandemic. In particular, among women who reported improvement or worsening due to the COVID-19 pandemic, as reported in Figure 1, waiting times, appointment hours, and accessibility were perceived as worsening, while most participants perceived cleaning and relational quality as improving.

At the end of the interview, the participants were asked to report their needs through an open-ended question. In total, 323 women did not respond to the survey. Moreover, some respondents (*n* = 722) provided one or more suggestions: 327 (45%) declared that it was necessary to have a specialised centre for endometriosis, 20% (186/722) requested more information about endometriosis disease and management, and 17% (145/722) needed economic support (Figure 2).

## 4. Discussion

This study aimed to investigate the frequency of complications, models of assistance, healthcare satisfaction, and QoL of a large sample of women who requested a visit for endometriosis in the 12 months prior to the interview, highlighting many relevant aspects of the health status of women with endometriosis in Italy during the COVID-19 pandemic. To our knowledge, this is the first study to evaluate the quality of care provided to women with endometriosis in Italy, demonstrating a high prevalence rate of complications and other diseases in the participants and the impact of QoL and other factors such as age, level of education, complications, and type of health service (specialised centre or doctor vs. non specialised centre) on different aspects of patient satisfaction.

Approximately 10% of the participants had a low level of education. For this segment of the population, it would be useful to reflect on the need for adequate communication and health education strategies. Higher education is linked to better health literacy, which is a crucial aspect especially in managing chronic diseases, as it can impact the selection of appropriate health services [34].

Regarding unhealthy behaviours, approximately 58% of the sample reported being physically inactive, which is higher than the 33.8% among Italian females aged 18–69 (95%CI: 33.0–34.5). The proportion of smoking (tobacco) and alcohol habits (occasionally and mainly during meals) was 21% and 55%, respectively, aligning with national data on smoking, alcohol consumption, and sedentary lifestyle [35]. However, it remains unclear whether exercise can improve the symptoms associated with endometriosis [36].

A very important finding of this study is that although the population investigated was young (age range, 18–55 years), the prevalence rate of complications was very high (79%), and the percentage of other pathological conditions reported was also high (55%). Previous studies have reported that endometriosis is associated with pelvic pain, dysmenorrhoea, dyspareunia, infertility, and a high risk of obstetric and/or surgical complications [37,38,39]. Therefore, it is important to consider the negative effects of endometriosis-induced infertility on mental health. Specifically, some authors have found that women affected by endometriosis and infertility often report mood disorders, loss of control, higher vigilance towards the onset or worsening of symptoms, social isolation, and catastrophizing [40]. Another study showed that women with both endometriosis and infertility have a greater risk of developing anxiety and depression [41].

A recent meta-analysis showed that women who suffer from endometriosis have a higher risk of preterm birth (OR 1.63; 95% CI, 1.32–2.01), miscarriage (OR 1.75; 95% CI, 1.29–2.37), placenta previa (OR 3.03; 95% CI, 1.50–6.13), small for gestational age (OR 1.27; 95% CI, 1.03–1.57), and caesarean delivery (OR 1.57; 95% CI, 1.39–1.78) compared with healthy women [42].

Further, Teng et al., showed that women with endometriosis also had other functional and/or pathological diseases [43]. A nationwide population-based cohort study of 6076 patients with endometriosis reported a prevalence rate of 24% for hypertension and 28.7% for hyperlipidaemia [44], and a recent meta-analysis highlighted significant genetic correlations between endometriosis and inflammatory conditions (asthma and osteoarthritis), as well as correlations between endometriosis and 11 pain conditions (migraine, headache, back pain, chronic back pain, and multisite chronic pain) [45]. Thus, it is necessary to consider the negative effects of chronic conditions on physical and mental HR-QoL [46].

More than half of the respondents stated that a period longer than 5 years had passed from the onset of symptoms to the diagnosis of endometriosis; for 35% of them, it was 10 years or more. Our findings are in line with the literature regarding delays from symptom onset to diagnosis due to the complexity of symptoms or the waitlist for diagnosis from laparoscopy [47]. Moreover, a longer delay has been reported for single women who experience a mean delay of 15.81 years; notably, 18% of single women reported a more than 20-year delay in diagnosis [41].

A multicentre study published in 2011 by Nnoaham et al. reported that the average delay in endometriosis diagnosis was, on average, 6.7 years internationally. The longest delay was in Italy (10.7 years) and the shortest was in China (3.3 years) [48]. However, a subsequent study showed a greater delay among Arab ancestry in the United Arab Emirates (11.6 years) [41]. Several reasons for the delayed diagnosis of endometriosis have been identified. First, the non-specific symptoms related to endometriosis can be confused with those of other typical gastrointestinal or urological diseases. The characteristics of this disease are largely unknown. Secondly, patient embarrassment and reliance on inadequate diagnostic methods must be considered. This diagnostic delay increases healthcare costs significantly more than an early diagnosis [49]. Furthermore, detention during diagnosis may include increased symptoms and disease severity, exacerbation of physical and psychosocial sequelae, and late access to effective treatment and care [50]. Therefore, patients with suspected endometriosis should be referred to multidisciplinary diagnostic-therapeutic units (gynaecologists, radiologists, general surgeons, psychologists, and urologists) with all the specific skills to arrive at the diagnosis quickly.

The two summary components of the SF-36 QoL scale analysed in our study revealed that women perceived higher levels of physical health status than mental health status: PCS = 38.7 (DS = 10.5) and MCS = 34.2 (11.2). As previously reported, the literature clearly shows the negative repercussions of endometriosis on mental health, specifically in pain management. Anxiety, depression, and pain catastrophizing are often reported as comorbid factors associated with endometriosis [12]. A randomised controlled study evaluated the role of psychological interventions in the management of chronic pelvic pain and QoL improvement. Interestingly, the authors found that in comparison to the waiting list, the groups following two different types of psychological intervention did not significantly reduce pain but significantly improved the QoL subscales for “control and powerlessness”, “emotional well-being”, and “social support”, as well as for endometriosis-related symptoms [51], demonstrating that psychological intervention can improve QoL beyond pain perception.

For the vast majority of patients, the main providers of care are specialised endometriosis centres or gynaecologists specialising in the treatment of endometriosis. Specifically, 6% declared that their family doctor mainly takes care of them for endometriosis, 4% reported a hospital, and 1% declared “other”. As reported above, multidisciplinary diagnostic and therapeutic units (gynaecologists, radiologists, general surgeons, psychologists, and urologists) can guarantee the best care, and family doctors or other professionals should refer women with suspected endometriosis to these units, as centres/networks of excellence are the only ways to ensure adequate care for women with endometriosis using a multidisciplinary approach. Clinical excellence can be achieved through appropriate training and adherence to evidence-based guidelines [52].

The perception of the quality of assistance received appeared to be more favourable among respondents. The overwhelming majority of those interviewed believed that the clinics they visited were readily accessible, maintained cleanliness, and offered satisfactory visitation hours. In fact, the proportions expressing positive satisfaction were 86% for visitation hours and 95% for both accessibility and cleanliness.

With regard to the dimension of perceived environmental quality (Table 3), the multivariable analysis revealed that positive satisfaction was not correlated with sociodemographic factors. Specifically, positive satisfaction was associated with higher levels of physical and psychological health status (PCS: AOR = 1.02; 95%CI: 1.00–1.04; *p* = 0.034; MCS: AOR: 1.03; 95%CI: 1.01–1.05; *p* = 0.001) in terms of visitation hours, the PCS component AOR = 1.05 (95%CI: 1.01–1.09; *p* = 0.006) for accessibility, the type of health provider (specialised vs non-specialised AOR = 3.46; 95%CI: 1.85–6.48; *p* < 0.001), and the presence of complications for the cleanliness question. Women with complications were found to be less satisfied than those without complications, with an AOR of 0.32; 95%CI: 0.12–0.85; *p* = 0.022.

Previous research has shown that medical complications can strongly impact patient satisfaction and that respondents with a perceived higher level of health status reported higher levels of satisfaction with healthcare [53].

In this study, the proportion of positive perceptions of relational satisfaction was lower than that of environmental quality. Specifically, participants evaluated healthcare professionals as courteous and helpful in 85% of instances (95%CI: 83–88), prepared to listen and clear in their explanations in 81% of instances (95%CI: 75–83), and available to listen in 73% of instances (95%CI: 70–76).

Among the independent factors investigated, the multivariate models highlighted that all sub-dimensions of relational satisfaction were significantly associated with physical and mental health status. In particular, this study demonstrated that women treated by specialists in endometriosis were more satisfied with the clarity of information communication (AOR = 1.80; *p* = 0.004; 95%CI: 1.20–2.69), and older participants were more satisfied with the availability to listen, as were women with complications. However, the primary critique concerned inadequate communication and cooperation among health professionals, with the proportion of satisfied women being 55% (95%CI: 51–58). As revealed by multivariable analysis, this dimension was related to educational level. Women with a higher educational level were satisfied (AOR = 0.62; 95%CI: 0.46–0.84; *p* = 0.002), as were women treated by specialists in endometriosis (AOR = 1.85; 95%CI: 1.30–2.63; *p* = 0.001), and this was significantly correlated with better physical and mental health status. It is plausible to suggest that the comprehension of the communication level, which is also influenced by educational attainment, may determine overall satisfaction with treatment. This finding underscores the need for medical staff to communicate in accordance with the patient’s educational level to ensure that all diagnostic and treatment information is accessible and comprehensible to all patients. Our findings align with other studies showing that communication with the patient, waiting time, age, perceived health status, and patient education affect patient satisfaction [54].

A recent survey carried out in France showed that among the factors associated with patient satisfaction, active listening took into account the patients’ opinions to prescribe the treatment and access to a coordinated model of care concerning the necessity of a multidisciplinary team. Specifically, patients with chronic diseases reported a need to be more actively involved in healthcare management [20].

More than half of the participants reported changes due to the pandemic in terms of organisation, environment, and relationships. Analysing the specific items, the perceived changes in environmental quality were in line with data from a recent meta-analysis [28].

Interestingly, the improvement was related to relational quality. The relationship between a patient’s health status and their satisfaction with healthcare personnel is crucial to the healthcare experience. Positive interactions and effective communication between patients and healthcare providers foster trust, engagement, and a better understanding of the patients’ needs, which can lead to improved health outcomes and higher satisfaction with the healthcare experience. In contrast, negative experiences and poor communication may lead to dissatisfaction and potentially hinder patients’ willingness to seek timely medical attention or adhere to treatment plans, which can have negative implications for their health status. Among the above-mentioned factors, effective communication might improve the quality of the therapeutic alliance and general patient satisfaction. Specifically, some authors have established that effective communication should provide patients with appropriate knowledge, clear information, mutual understanding, respect, and empathy. However, specialist equipment should adopt strategic measures to promote effective communication, such as patient communication needs, mentorship, reflection, and education in practice [55].

Several factors may contribute to the physician-patient relationship, and understanding the patient’s perspective can help improve patient outcomes and overall healthcare quality. The patients’ point of view represents the major strength of this study in investigating the assistance of women diagnosed with endometriosis in Italy during the COVID-19 pandemic, their satisfaction, and their quality of life. However, to better understand the patient’s perspective, it is fundamental to consider their unmet needs. Satisfaction of unmet needs can improve patient satisfaction with health services. Some authors have highlighted the main areas of unmet needs in women with endometriosis, such as emotional health, social support, looking after and caring for one’s body, patient empowerment, interpersonal issues, general endometriosis information, and physical health [56].

It is important to note that the population included in this study may not be fully representative of all women with endometriosis in Italy. Additionally, the online survey format might have excluded women who do not use social networks or messaging applications. Nonetheless, it may represent an attempt to establish a relationship with the population of interest by considering the advantages of online survey research [57].

Second, the data analysed were self-reported and not clinical, and we did not collect information about the severity of endometriosis.

In addition, as highlighted by the WHO, although several screening tools exist for the specific measurement of endometriosis, none is currently validated to accurately identify or predict individuals or populations that are most likely to have the disease. This evidence further directed us towards the choice of tools that could measure constructs transversal to different diseases. Furthermore, the results should also be interpreted according to the reference healthcare system. For example, previous studies on diagnostic delay have reported that data collected outside Italy focused on the role of the national health system and underlined how the diagnostic delay is much longer in countries that are predominantly state-funded compared with those that are self- or insurance-funded [48].

Hence, our data on the unmet needs of women with endometriosis partly agree with those in the literature. Italian women referred to the need for specialised centres for endometriosis to obtain more precise information about the disease and its management and to receive a better quality of relationship with specialised equipment. This probably shows the quality of Italian services, which evidently lack specialised support.

Finally, the results are primarily applicable to women who have had a recent medical visit related to endometriosis within the past year, specifically those who are recently diagnosed or whose endometriosis is not effectively treated. It is important to note that well-treated or long-standing cases of endometriosis in older women may not have been included in this analysis.

## 5. Conclusions

As reported by many authors, the complexity of the diagnosis, treatment, and management of endometriosis requires important efforts, including economic ones, on the part of patients, their families, and society. The results presented allow us to emphasize some aspects. First of all, the importance of specialised centres since patients treated in specialised centres reported higher satisfaction levels, particularly in the clarity of information provided. This emphasizes the need for healthcare professionals to prioritize clear and accessible communication about the disease, treatment options, and management strategies. Second, a substantial portion of participants reported being sedentary and having a higher BMI. In fact, research has established a clear association between excess body weight and the severity of endometriosis symptoms. Higher BMI levels have been linked to increased inflammation, hormonal imbalances, and elevated levels of oestrogen, all of which can exacerbate the progression of endometriosis. Moreover, excess adipose tissue can secrete pro-inflammatory cytokines, further contributing to pain and discomfort. This indicates a need for incorporating lifestyle interventions and weight management strategies as part of the comprehensive care plan for endometriosis patients. Third, understanding patient preferences and needs regarding medication is essential. While some prefer to take medicines only if prescribed by specialists, a significant number admitted to using painkillers without specialist recommendation. Tailoring medication management plans to individual patient preferences and needs is vital. Fourth, a large majority of participants reported experiencing complications related to endometriosis. This highlights the importance of adopting a comprehensive approach to managing complications, with a focus on addressing specific issues like dyspareunia and chronic pelvic pain. Finally, this study indicates that participants perceived their physical well-being as better than their mental well-being. This suggests the need for an integrated approach to address both physical and mental health aspects in endometriosis care. From a clinical point of view, in fact, the results obtained critically underline the importance of referring patients with chronic pelvic pain immediately to multidisciplinary teams.

Our contribution aligns with the literature dedicated to endometriosis, prompting a significant reflection on patient needs and advocating for further research from the patient’s standpoint. This approach also considers the COVID-19 pandemic’s impact on healthcare systems.

## Figures and Tables

**Figure 1 ijerph-20-06978-f001:**
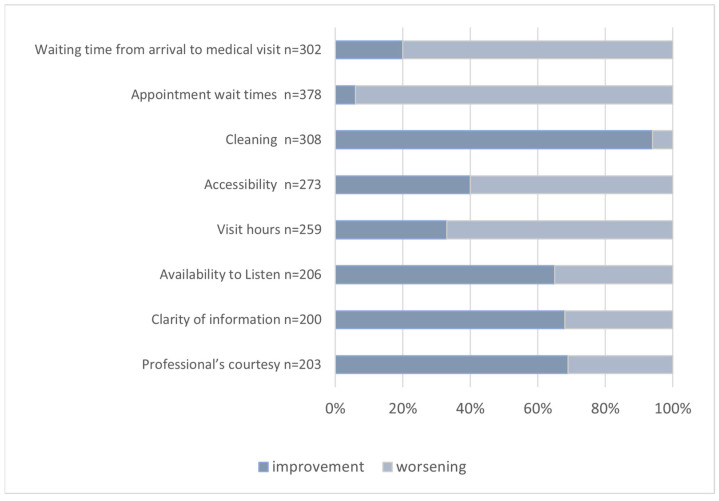
Perceived Improvement or Worsening Related to COVID-19.

**Figure 2 ijerph-20-06978-f002:**
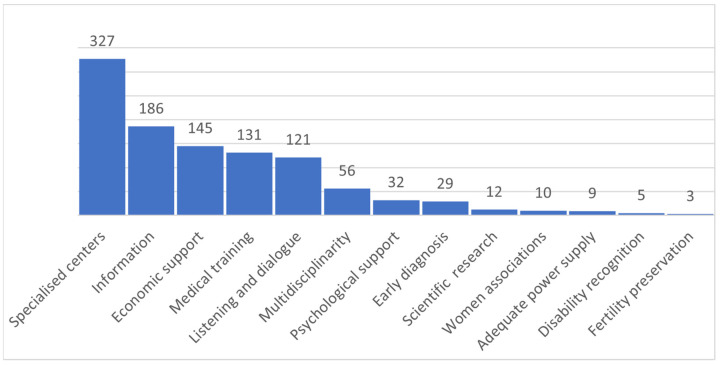
Needs of the participants (one or more than one answer).

**Table 1 ijerph-20-06978-t001:** Characteristics of the sample *n* = 875.

Characteristics	Mean (SD) or N (%)
Age	34 (7.6)
Residence	
- South and Islands	320 (37)
- Centre	187 (21)
- North	368 (42)
Nationality	
- Italian	857 (98)
- Foreign	18 (2)
Marital status	
- Married	271 (40)
- Cohabiting	227 (26)
- Single	346 (31)
- Widower	2 (0)
- Divorced	29 (3)
Living alone	
- No	776 (89)
- Yes	99 (11)
Educational qualification	
- Postgraduate	113 (13)
- Degree	286 (33)
- Diploma	390 (44)
- Middle school certificate	85 (10)
- Elementary school certificate	1 (0)
Do you currently have a job?	
- Yes, indefinitely	420 (48)
- Yes, for a fixed term	155 (18)
- No, I am unemployed	160 (18)
- No, I am a student	89 (10)
- No, I am a housewife	51 (6)
- No, I am retired	0 (0)
Body Mass Index, categories (kg/m^2^)	
- Underweight (<18.5)	112 (13)
- Normal weight (18.5–24.9)	526 (61)
- Overweight (25.0–29.9)	156 (18)
- Obese (≥30.0)	72 (8)
Do you do physical activity?	
- Yes, moderate physical activity	335 (39)
- Yes, intense physical activity	29 (3)
- No, I am not currently physically active	484 (55)
- I have never exercised in my life	27 (3)
Smoking habit: currently you:	
- Yes, I smoke cigarettes (tobacco)	183 (21)
- Yes, I smoke an electronic cigarette	47 (5)
- No, but I have smoked in the past	214 (25)
- No, I have never smoked	431 (49)
Do you ever drink a unit of alcohol (1 glass of beer or 1 glass of wine or spirits)?	
- Yes, occasionally	373 (43)
- Yes, mainly during meals	101 (12)
- Yes, only during meals	71 (8)
- Yes, mainly out of meals	38 (4)
- Yes, only out of meals	9 (1)
- I do not drink alcohol	283 (32)
Duration of the disease (years)	
- 0–3	478 (55)
- 4–9	223 (25)
- >9	174 (20)
Time from symptoms to diagnosis (years)	
- <1	167 (19)
- 1–4	208 (24)
- 5–9	197 (22)
- ≥10	303 (35)
Hormonal treatments (lifetime perspective)	
- Yes	598 (68)
- No	277 (32)
Therapies (Do you take other medicines not prescribed or recommended by a doctor for problems related to endometriosis?)	
- Yes, painkillers	360 (41)
- Yes, supplements	192 (22)
- Yes, homeopathic products	8 (1)
- Yes, other	10 (1)
- No, I only take the medicines prescribed or recommended by my gynaecologist	187 (21)
- No, I am not taking any medications	118 (14)
Clinical complications related to endometriosis	
- Yes	687 (79)
- No	188 (21)
Surgical intervention for endometriosis and/or complications *	
- Yes	527 (60)
- No	348 (40)
Complications	
- Pelvic pain	305 (34.9)
- Dyspareunia	598 (68.3)
- Pelvic floor disorders	443 (50.6)
- Self catheterism	34 (3.9)
- Neuropathic disorder	273 (31.2)
- Infertility	316 (36.1)
- Intestinal stenosis	101 (11.5)
- Adherence	489 (55.9)
Comorbidities **	
- Yes	485 (55)
- No	390 (45)
Health Status (SF-36 score)	
- Physical Summary Component (PCS)	38.7 (10.5)
- Mental Summary Component (MCS)	34.2 (11.2)

* complications: at least one complication among pelvic pain, dyspareunia, pelvic floor disorder, self catheterism, neuropathic disorder, infertility, hysterectomy, salpingectomy, ovariectomy, intestinal stenosis, intestinal resection, bladder resection, adherence, other. ** comorbidities were counted from the item “do you suffer from other diseases” (yes/no)?

**Table 2 ijerph-20-06978-t002:** Health structure or professionals.

**Centres/Hospitals/Professionals Chosen for the Treatment of One’s Illness**	***N* (%)**
Centre or doctor specialised in endometriosis treatment	698 (80)
Gynaecologist	75 (9)
Family Doctor	54 (6)
Hospital	35 (4)
Other	13 (1)
**Organizational Quality of centre/hospital/clinic visited**	***N* (%)**
Appointment waits times (≤6 months)	871 (88)
Waiting time from arrival to medical visit (≤30 min)	585 (67)
Appointment scheduled (yes)	174 (20)
**Healthcare professionals consulted during the last year for complications related to endometriosis**	***N* (%)**
Endocrinologist (yes)	131 (20)
Gastroenterologist (yes)	189 (28)
Proctologist (yes)	71 (11)
Urologist (yes)	90 (13)
Radiologist (yes)	247 (37)
Pain specialist (pain therapy) (yes)	44 (7)
Physiatrist (yes)	56 (8)
Physiotherapist (yes)	183 (27)
Sexologist (yes)	21 (3)
Psychologist (yes)	155 (23)
Nutritionist (yes)	299 (45)
Psychiatrist (yes)	38 (6)

**Table 3 ijerph-20-06978-t003:** Environment Quality perceived (positive perceptions). Multivariable analysis: visit hours, accessibility, and cleaning dependent variables (N = 875).

**Dependent: Visit Hours *n*, %, [95%CI]** **751, 86%, [83–88]**	**OR**	** *p* ** **-Value**	**95%CI**
Age	1.00	0.692	0.98–1.03
Residence (rif: Southern Italy)			
Central Italy	1.56	0.117	0.89–2.73
Northern Italy	1.13	0.575	0.73–1.76
Occupational status (rif: employed)	0.93	0.759	0.60–1.45
Educational level (rif: high)	0.70	0.092	0.44–1.06
Marital status (rif: single)	1.24	0.315	0.82–1.87
Health Service (rif: non-specialised centre or doctor)	1.56	0.058	0.98–2.46
Comorbidities (rif: no)	0.67	0.066	0.44–1.03
Complications (rif: no)	0.69	0.150	0.41–1.14
PCS	1.02	0.034	1.00–1.04
MCS	1.03	0.001	1.01–1.05
**Dependent: Accessibility *n*, %, [95%CI]** **831, 95%, [93–96]**	**OR**	** *p-* ** **Value**	**95%CI**
Age	1.02	0.308	0.98–1.07
Residence (rif: Southern Italy)			
Central Italy	1.17	0.702	0.53–2.58
Northern Italy	1.93	0.76	0.93–4.02
Occupational status (rif: employed)	0.96	0.904	0.48–1.90
Educational level (rif: high)	0.89	0.738	0.46–1.74
Marital status (rif: single)	1.16	0.652	0.60–2.24
Health Service (rif: non-specialised centre or doctor)	1.57	0.213	0.77–3.18
Comorbidities (rif: no)	0.73	0.366	0.38–1.43
Complications (rif: no)	0.39	0.054	0.15–1.02
PCS	1.05	0.006	1.01–1.08
MCS	0.99	0.519	0.96–1.02
**Dependent: Cleaning *n*, %, [95%CI]** **828, 95%, [93–96]**	**OR**	** *p* ** **-Value**	**95%CI**
Age	0.97	0.179	0.93–1.01
Residence (rif: Southern Italy)			
Central Italy	0.78	0.504	0.38–1.60
Northern Italy	1.99	0.076	0.93–4.26
Occupational status (rif: employed)	0.83	0.599	0.41–1.67
Educational level (rif: high)	1.08	0.821	0.57–2.05
Marital status (rif: single)	1.13	0.715	0.59–2.13
Health Service (rif: non-specialised centre or doctor)	3.46	<0.001	1.85–6.47
Comorbidities (rif: no)	0.98	0.959	0.53–1.54
Complications (rif: no)	0.32	0.022	0.12–0.85
PCS	1.00	0.930	0.97–1.03
MCS	1.02	0.091	1.00–1.05

**Table 4 ijerph-20-06978-t004:** Analysis of Perceived Relational Quality (Positive Perceptions) with Courtesy, Clarity of Information, and Availability to Listen as Dependent Variables.

**Dependent: Professional’s Courtesy *n*, %, [95%CI]** **748, 85%, [83–88]**	**OR**	** *p* ** **-Value**	**95%CI**
Age	1.00	0.823	0.97–1.03
Residence (rif: Southern Italy)			
Central Italy	0.62	0.071	0.34–1.01
Northern Italy	0.71	0.153	0.45–1.13
Occupational status (rif: employed)	1.14	0.535	0.74–1.76
Educational level (rif: high)	0.72	0.113	0.48–1.08
Marital status (rif: single)	0.96	0.838	0.63–1.45
Health Service (rif: non-specialised centre or doctor)	1.47	0.096	0.93–2.33
Comorbidities (rif: no)	0.94	0.789	0.63–1.42
Complications (rif: no)	1.18	0.487	0.74–1.87
PCS	1.03	0.015	1.01–10.5
MCS	1.04	<0.001	1.02–1.06
**Dependent: Clarity of information n, %, [95%CI]** **707, 81%, [78–83]**	**OR**	** *p* ** **-Value**	**95%CI**
Age	1.02	0.132	0.99–1.04
Residence (rif: Southern Italy)			
Central Italy	1.26	0.349	0.78–2.04
Northern Italy	1.12	0.573	0.75–1.67
Occupational status (rif: employed)	0.91	0.659	0.62–1.35
Educational level (rif: high)	1.06	0.736	0.73–1.54
Marital status (rif: single)	1.67	0.413	0.80–1.69
Health Service (rif: non-specialised centre or doctor)	1.80	0.004	1.20–2.69
Comorbidities (rif: no)	0.91	0.609	0.63–1.31
Complications (rif: no)	1.16	0.471	0.77–1.77
PCS	1.03	<0.001	1.02–1.06
MCS	1.03	<0.001	1.02–1.05
**Dependent: Availability to Listen n, %, [95%CI]** **640, 73%, [70–76]**	**OR**	** *p* ** **-Value**	**95%CI**
Age	1.02	0.037	1.00–1.05
Residence (rif: Southern Italy)			
Central Italy	0.94	0.760	0.61–1.43
Northern Italy	1.02	0.909	0.71–1.47
Occupational status (rif: Unemployed)	1.07	0.686	0.76–1.52
Educational level (rif: low)	0.82	0.248	0.59–1.14
Marital status (rif: single)	0.78	0.159	0.56–1.10
Health Service (rif: non-specialised centre or doctor)	1.42	0.067	0.97–2.07
Comorbidities (rif: no)	0.83	0.257	0.59–1.15
Complications (rif: no)	1.46	0.044	1.01–2.11
PCS	1.03	<0.001	1.02–1.05
MCS	1.03	<0.001	1.02–1.05

**Table 5 ijerph-20-06978-t005:** Analysis of Satisfaction with Cooperation Among Professionals (Positive Perceptions) with Cooperation as the Dependent Variable.

Dependent: Cooperation *n*, %, [95%CI] 480, 55%, [51–58]	OR	*p*-Value	95%CI
Age	0.99	0.349	0.97–1.01
Residence (rif: Southern Italy)			
Central Italy	0.95	0.798	0.65–1.39
Northern Italy	0.97	0.839	0.70–1.33
Occupational status (rif: employed)	0.93	0.652	0.68–1.28
Educational level (rif: low)	0.62	0.002	0.46–0.84
Marital status (rif: single)	1.04	0.769	0.78–1.41
Health Service (rif: non-specialised centre or doctor)	1.85	0.001	1.30–2.63
Comorbidities (rif: no)	0.88	0.384	0.65–1.18
Complications (rif: no)	1.03	0.881	0.73–1.44
PCS	1.03	<0.001	1.01–1.04
MCS	1.03	<0.001	1.02–1.05

## Data Availability

The data analysed in this study are not available to outside researchers due to privacy issues.

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
