# Peer review of "The Perception of the Quality of Professional Healthcare Assistance for the Management of Endometriosis: Findings from a National Survey in Italy"

_ijerph, 2023, doi:10.3390/ijerph20216978_

Round 1

Reviewer 1 Report

this interesting paper offers to the reader a glance of the italian situation in a subgroup of women affected by endometriosis.

the authors should comment on:

choice of survey (why not something more specific for endometriosis)

comparison and comment on similar data collected outside Italy, aiming if possible to evaluate the role of a national health system

minor revision    is recommended

Reviewer 2 Report

The Perception of the Quality of Professional Health Care Assistance for the Management of Endometriosis. Findings from a National Survey in Italy

Endometriosis is a chronic, benign gynecological disorder affecting a significant percentage of women, especially of reproductive age. Diagnosing endometriosis remains difficult because of the variety of symptoms and indicators, which can also cause it to be mistaken for other disorders. The health burden of women with endometriosis is significant. The authors analyzed patients’ points of view regarding the quality of care for endometriosis during the COVID-19 pandemic and in particular, the quality of assistance received during the COVID-19 pandemic and its association with the management of the endometriosis, the perceived mental and physical well-being, and the perceived changes in the assistance due to the COVID-19 pandemic, which is clinical relevant and very interesting. This information could guide the debate, organization of health care and further research on this topic.

Title: The title is well chosen, reflecting the study being reported.

Overall:

The aim is well emphasized and explained. The paper is well written and was very pleasant to read.

Abstract:

No comments.

1.      Introduction :

The introduction section is attractive to read, emphasizing the reason for conducting the study, explaining the background of the different possible uterine abnormalities.

No comment on this section.

2.      Materials and methods

This section is well written, explaining the material and methods of this study.

No comment on this section.

3.      Results

This section is well written and very pleasant to read. Despite some points need explanation and or clarification.

1.      On page 4 line 181-183 the authors report the following: “ Based on the BMI data, 181 26% were classified as overweight or obese, and more than half (58%) reported being sedentary. “ This sentence is not clearly for the reader, what is meant with being sedentary? Was the BMI normal? Please explain and re[ort in an understandable fashion.

2.      The authors report a couple of outcomes on page 4  ( line 187-193)  which are not re4ported in table 1. For a good overview I want to urge the authors to add these results to table 1.

3.      As a reader, I was curious if the result presented account for women treated in a specialized center as well as women not treated in a specialized center. This is crucial and important information to know. Do women in specialized center received more treatment, are the outcomes (including quality if life better). From a clinical view very relevant question which could be answered by the authors on basis of the available information.

4.      Discussion

This section is well written and very pleasant to read.

4.     The authors could elucidate what the presented results could mean for clinical practice. What are the lessons learned on basis of the current presented results?

The English grammar is excellent overall, little mistakes encountered.
